



# Benefits of a second tandem flight phase between two successive satellite altimetry missions for assessing the instrumental stability

Michaël Ablain[1], Noémie Lalau[1], Benoit Meyssignac[2], Robin Fraudeau[1], Anne Barnoud[1], Gérald Dibarboure[3], Alejandro Egido[4], and Craig Donlon[4]

[1]MAGELLIUM, Ramonville Saint-Agne, 31520, France
[2]LEGOS, CNES, CNRS, IRD, Université Paul Sabatier, Toulouse, 31400, France
[3]CNES, Toulouse, 31400, France
[4]ESA, ESTEC, Noordwijk, 2201 AZ, The Netherlands

**Correspondence:** Michaël Ablain (michael.ablain@magellium.fr)

**Abstract.**

The five successive reference missions, TOPEX/Poseidon, Jason-1, Jason-2, Jason-3, and more recently Sentinel-6 Michael Freilich, have ensured the continuity and stability of the altimetry data record. Tandem flight phases have played a key role in verifying and ensuring the consistency of sea level measurements between successive altimetry reference missions and thus the stability of sea level measurements. During a tandem flight phase, two successive reference missions follow each other on an identical ground track at intervals of less than one minute. Observing the same ocean zone simultaneously, the differences in sea level measurements between the two altimetry missions mainly reflect their relative errors. Relative errors are due to instrumental differences related to altimeter characteristics (e.g., altimeter noise) and processing of altimeter measurements (e.g., retracking algorithm), precise orbit determination, and mean sea surface. Accurate determination of systematic instrumental differences is achievable by averaging these relative errors over periods that exceed 100 days. This enables for the precise calibration of the two altimeters. The global mean sea level offset between successive altimetry missions can be accurately estimated with an uncertainty of about ± 0.5 mm ([16-84]% confidence level). Nevertheless, it is only feasible to detect instrumental drifts in the global mean sea level exceeding 1.0 to 1.5 mm per year, due to the brief duration of the tandem phase (9 to 12 months). This study aims to propose a novel cross-validation method with a better ability to assess the instrumental stability (i.e. instrumental drifts in the global mean sea level trends). It is based on the implementation of a second tandem flight phase between two successive satellites a few years after the first one. Calculating sea level differences during the second tandem phase provides an accurate evaluation of relative errors between the two successive altimetry missions . With a second tandem phase long enough, the systematic instrumental differences in sea level will be accurately reevaluated. The idea is to calculate the trend between the systematic instrumental differences made during the two tandem phases. The uncertainty in the trend is influenced by the length of each tandem phase and the time intervals between the two tandem phases. Our findings show that assessing the instrumental stability with two tandem phases can achieve an uncertainty below ±0.1 mm yr$^{-1}$ ([16-84]% confidence level) at the global scale , for time intervals between the two tandem phases higher than 4 years or more, and each tandem phase lasts at least four months. On regional scales, the gain is greater with an uncertainty of ±0.5 mm yr$^{-1}$ ([16-84]% confidence level) for spatial scales of about 1000 km or more. With regard to the scenario foreseen for the





second phase between Jason-3 and Sentinel-6 Michael Freilich planned for early 2025, 2 years and 9 months after the end of the first tandem phase, the instrumental stability could be assessed with an uncertainty of $\pm 0.14$ mm yr$^{-1}$ on the global scale, and $\pm 0.65$ mm yr$^{-1}$ for spatial scales of about 1000 km ([16-84]% confidence level). In order to take a larger benefit of this novel cross-validation method, this involves regularly implementing double tandem phases between two successive altimetry missions in the future.

## 1  Introduction

The sea level data record has been continuously calculated using multiple satellite altimetry missions since January 1993. The five successive reference missions, TOPEX/Poseidon (TP), Jason-1, Jason-2, Jason-3, and more recently Sentinel-6 Michael Freilich (S6-MF), have ensured the continuity and stability of the sea level data record. The upcoming launches of the Coperni­cus Sentinel-6B and Sentinel-6C satellites in the next decade are expected to maintain this continuity and stability. It is crucial

to maintain the stability of the sea level data record to effectively monitor the effects of current climate change, including sea level rise and acceleration ((Meyssignac et al., 2023)), as well as oceanic heat uptake and the Earth energy imbalance ((Hakuba et al., 2021; Marti et al., 2022, 2023))

The main sources of uncertainties in the stability of the sea level data record have been identified and characterised by (Ablain et al., 2019b; Guérou et al., 2023; Prandi et al., 2021). First, they are attributed to short-term time-correlated errors (< 1 year)

in altimeter measurements (altimeter range, altimeter-related corrections such as sea state bias and ionosphere effects), precise orbit determination (POD), geophysical, and atmospheric corrections. Second, they arise from long-term time-correlated errors (> 5 years) in POD, in wet troposphere correction, and glacial isostatic adjustment (GIA). They are also related to uncertainties in the estimate of the sea level offset between two successive reference missions. These different sources of uncertainties reveal an uncertainty in the trend of global mean sea level (GMSL) around $\pm$ 0.7 mm yr$^{-1}$ ([5-95]% confidence level, CL) over a 10-

45  year period and down to $\pm$ 0.4 mm yr$^{-1}$ ([5-95]% CL) over a 20-year period and beyond (Guérou et al., 2023). The uncertainty in the acceleration of the GMSL is estimated to be close to 0.07 mm yr$^{-2}$ over a 25-year period ([5-95]% CL) (Guérou et al., 2023). On a regional scale of a few hundred kilometres, uncertainties in mean sea level trends range from 0.7 to 1.3 mm yr$^{-1}$ ([5-95]% CL) (Prandi et al., 2021). This stability performance exceeds the requirements of altimetry missions (Donlon et al., 2021).

Tandem flight phases (hereafter named "tandem phase") have played a key role in verifying and ensuring the consistency of sea level measurements between successive reference missions. The tandem phases have been implemented after the launch of each new reference mission: TP and Jason-1 (2002), Jason-1 and Jason-2 (2008), Jason-2 and Jason-3 (2016), and Jason-3 and S6-MF (2021-2022). During a tandem phase, the two successive reference missions follow each other on an identical ground track at intervals of less than one minute. Upon completion of each tandem phase, the older reference mission is moved over

an alternative orbit to enhance sea level observations (see, e.g., (Dibarboure et al., 2012)).

Donlon et al. (2016, 2020, 2023) clearly explain the scientific justification and mission benefits of the tandem phase based on the recommendations of the Global Climate Observing System (GCOS). During a tandem phase, we can reasonably assume





that the ocean and atmosphere at the scales we are interested in do not vary significantly between measurements made by the two altimetry missions. Therefore, by comparing the sea level measurements from the two altimeters, the geophysical and

atmospherical effects are cancelled. We can therefore accurately determine the relative errors made by both altimetry missions. The relative errors are made up of several effects due to: a) instrumental differences related to the altimeter characteristics (e.g. altimeter noise, how a radar pulse interacts with the ocean surface, e.g. Dibarboure et al. (2014)) and the processing of the altimeter measurements (e.g. algorithm retracking, sea state bias correction); b) differences in the precise orbit determination (POD), Couhert et al. (2015); Rudenko et al. (2023); c) differences in the Mean Sea Surface (MSS) in areas of strong geoid

gradients, e.g. (Schaeffer et al., 2023) because the two satellites are not exactly on the ground track ($\pm$ 1 km). The systematic instrumental differences are obtained after averaging the short-term time-correlated effects on the relative errors (e.g., instrumental noise, differences in the POD and the MSS) over a period of at least 100 days ((noa, 2008)). Tandem phases allowed the precise calibration of the two altimeters, with the detection of systematic instrumental differences of a few millimetres to a few centimetres at different spatial scales from a few hundred kilometres to the global scale (for example, (Dorandeu et al.,

2004; Ablain et al., 2010; Cadier et al., 2024)). They allowed in particular the accurate estimation of the GMSL offset between two successive altimetry missions on the order of a few millimetres to a few centimetres (depending on the altimetry missions). The low level of uncertainty in the GMSL offset, close to $\pm$ 0.5 mm ([16-84]%) (Ablain et al., 2019b), allows us to accurately link the GMSL time series and thus reduce the uncertainty in the GMSL trend by less than $\pm$ 0.05 mm yr$^{-1}$ ([16-84]% CL) over a 10-year period (Zawadzki and Ablain (2016)).

However, it is only feasible to detect instrumental drifts in the global mean sea level exceeding 1.0 to 1.5 mm per year, due to the brief duration of the tandem phase (9 to 12 months). Other methods have been developed and used for more than 30 years to verify the long-term stability of altimeter measurements, within the scope of altimetry validation activities. They are based on cross-comparisons with altimetry missions together (e.g., crossover comparison, along-track comparison), on comparisons with independent measurements (e.g., ocean model reanalysis, in situ data such as tide gauge measurements), and also on the

assessment of the sea level budget closure (Dieng et al., 2017; Barnoud et al., 2021). Each validation method has a specific uncertainty and the potential to detect drift in the sea level data record. For example, comparison of altimetry measurements and tide gauge data allows the detection of a trend in GMSL differences with an uncertainty of approximately $\pm$ 0.7 mm yr$^{-1}$ ([5-95]% CL) over a 10-year period (Ablain et al., 2018; Watson et al., 2021). An instrumental drift on the TOPEX-A GMSL of about 1.5 mm yr$^{-1}$ from 1993 to 1999 was detected with this method (Watson et al., 2015; Ablain et al., 2018). Another

example is the direct comparison of the along-track sea level measurements between two altimetry missions on different orbits. It allows the detection of trends in sea level differences with an uncertainty of about $\pm 0.3$ m$^{-1}$ ([5-95]% CL) on the global scale and $\pm 1.2$ mm $^{-1}$ ([5-95]% CL) at regional scales over a 10-year period (Jugier et al., 2022). A drift in the Sentinel-3A GMSL of approximately 1.2 mm yr$^{-1}$ $\pm 0.6$ m$^{-1}$ ([5-95]% CL) from 2016 to 2021, was detected due to an error in processing altimeter measurements (Jugier et al., 2022). For all of these methods, the ocean is not observed at the same location and/or at

the same time. As a consequence, sea level differences include part of the oceanic variability and additional errors (geophysical and atmospheric corrections, POD) not cancelled in sea level differences. Both of these effects limit our ability to detect a drift in the altimeter measurements. The situation would be different if the satellite altimeter system and commensurate in situ





fiducial measurement reference system were capable of fully sampling the same ocean variability (e.g. the same tides, waves, ocean dynamics, and their regional geographic patterns amongst other aspects).

This study aims to propose a novel cross-validation method with a better ability to assess instrumental stability. We propose the realisation of a second tandem phase between two successive altimetry missions to evaluate the relative stability of two successive altimetry missions. The fundamental concept is illustrated in Fig. 1. This entails relocating the previous altimetry mission back to its initial orbit several years after the first tandem phase. The two consecutive altimetry missions will once more be positioned less than a minute apart in identical orbit. With a second tandem phase long enough and 2 tandem phases

separated by a time long enough, the systematic instrumental differences between the two altimetry missions will be reassessed. We can then analyse how systematic instrumental differences have changed between the first and second tandem phases by calculating the trend of sea level differences over the period that includes both tandem phases (blue curve in Fig. 1).

In this paper, we demonstrate the ability of the 2-tandem phase method to assess instrumental stability. We explain the method developed to quantify the uncertainty of the 2-tandem phase method in sections 2 and 3. We assess the global uncertainty of the

2-tandem phase method, which varies based on the length of the second tandem phase and the interval between the two phases in Section 4 Additionally, we evaluate the method's uncertainty at regional scales by examining its sensitivity to spatial scales ranging from several hundred to one thousand kilometres. Finally, we compare the uncertainty obtained with other validation methods to highlight the benefits of the second tandem phase for assessing the instrumental stability in section 5. We also discussed the results with regard to the scenario foreseen for the second phase between Jason-3 and S6-MF (Ferrier, 2023)

planned for early 2025, 2 years and 9 months after the end of the first tandem phase.

## 2   Method to quantify the uncertainty of the 2-tandem phase method

The uncertainty of the two tandem phases is characterised by the uncertainty linked to the trend in sea level differences throughout the period covering both tandem phases (illustrated by the blue curve in Fig. 1). Our approach is based on the framework established in (Ablain et al., 2019b; Prandi et al., 2021) to assess uncertainties in trends at both global and regional

scales. First, it involves computing an uncertainty budget for sea level differences during a tandem phase. This uncertainty budget enumerates the various sources of uncertainty that impact the time series. In addition, it provides the temporal correlation and standard deviation associated with errors. Once the uncertainty budget is established (see the following section), the error covariance matrix for sea level differences observed during a tandem phase ($\Sigma_{tp}$, where tp stands for tandem phase) is derived using the methodology outlined in (Ablain et al., 2019b).

The second step consists of using this error covariance matrix ($\Sigma_{TP}$) to calculate the uncertainty of the trend in sea level differences. The formalism used is also based on (Ablain et al., 2019b). The trend is fitted from a linear regression model ($y = X\beta + \epsilon$) applying an ordinary least squares (OLS) approach where the estimator of $\beta$ with the OLS, noted $\hat{\beta}$, is:

$$\hat{\beta} \sim (X^t X)^{-1} X^t y \tag{1}$$



Where $X$ is the time vector that contains the date of each altimeter cycle during the two tandem phases, $y$ is the observation vector that contains the sea level differences averaged over each complete cycle. The uncertainty in the trend is given by $\hat{\beta}$, which is the distribution of the estimator following a normal law:

$$\hat{\beta} = N(\beta, (X^t X)^{-1}(X^t \Sigma X)(X^t X)^{-1}) \tag{2}$$

Here, $\Sigma$ is the error covariance matrix of the relative errors observed during the two tandem phases. $\Sigma$ is divided into two distinct diagonal blocks with the error covariance matrix during the first tandem phase ($\Sigma_{tp\_1}$) and during the second tandem phase ($\Sigma_{tp\_2}$), as shown below:

$$\Sigma = \begin{pmatrix} \Sigma_{tp\_1} & \begin{matrix} 0 & \dots & 0 \\ \vdots & \ddots & \vdots \\ 0 & \dots & 0 \end{matrix} \\ \hline \begin{matrix} 0 & \dots & 0 \\ \vdots & \ddots & \vdots \\ 0 & \dots & 0 \end{matrix} & \Sigma_{tp\_2} \end{pmatrix} \tag{3}$$

It should be noted that the calculation of trend uncertainty does not depend on sea level differences ($y$), but only depends on the time vector ($X$) and the error covariance matrix ($\Sigma$). Thus, we can study the uncertainty of the 2-tandem phase method without having yet executed the second tandem phase.

## 3   Calculation of the uncertainty budget during a tandem phase

### 3.1   Mean sea level differences calculation during the tandem phase

We calculate the sea level estimates over the 3 tandem phases between Jason-1 and Jason-2 (August 2008 - January 2009), between Jason-2 and Jason-3 (February 2016 - October 2016) and between Jason-3 and of S6-MF (September 2021 - April 2022). It should be mentioned that the tandem phase between Jason-3 and S6-MF started on 17 December 2020 and ended on 4 April 2022. However, an instrumental anomaly was detected on side A of the S6-MF altimeter (Poseidon-4) a few months after the launch of S6-MF. Thus, on 14 September 2021 a switch was operated on side B of the S6-MF altimeter (Dinardo et al., 2022). Therefore, the data of the first tandem phase between S6-MF and Jason-3 can be used after the side B change from 14 September 2021 to 4 April 2022. For Jason-1, Jason-2, and Jason-3, the altimeter products used are the non-time critical (NTC) along-track level-2+ (L2P) products from the Copernicus Marine Service under the CNES responsibility. These products are downloaded from the Aviso ftp website (http://ftp-access.aviso.altimetry.fr/uncross-calibrated/open-ocean/non-time-critical/l2p/sla/) and contain the along-track sea level anomaly at 1Hz (SLA, see Eq. (1)) calculated after applying a validation process fully described in the product handbook of each altimetry mission (Along-track Level-2+ (L2P) Sea Level





Anomaly Sentinel-3 / Jason-CS-Sentinel-6 Product Handbook, 2022). We use version 03_00 which is the current product version of Sentinel-3 and S6-MF L2P, with respective reprocessing from Baseline Collection 004 (PB2.61) and Baseline Collection F05. The along-track SLA provided in the L2P products is derived from the following equation:

$$SLA = Orbit - Range - \Sigma_i Correction_i - Mean\,Sea\,Surface$$

The geophysical corrections applied in L2P products for the SLA calculation are homogenised for each altimetry mission and identical during a tandem phase. We have also verified that whether or not geophysical and atmospheric corrections are applied, the sea level estimates remain consistent during a tandem phase. It should also be noted that the wet troposphere correction

derived from microwave radiometers has not been applied to the sea level calculation to avoid the introduction of instrumental errors not related to the altimeter measurements. The mean sea level is then calculated on regional scales by averaging the along-track SLA at 1 Hz in cells of 1 degree in latitude per 3 degrees in longitude, over a complete repetitive cycle ($\approx 9.91$ days for reference altimetry missions). The mean sea level differences are obtained on regional scales by comparing directly the previously calculated regional mean sea level. GMSL differences are obtained by averaging the mean sea level differences

at regional scales applying the GMSL AVISO method (Henry et al., 2014).

### 3.2   Uncertainty budget on the global scale

In the case of a tandem phase, the uncertainty budget of sea level differences is extremely simplified since most of the different sources of uncertainties described in the mean sea level uncertainty budget are cancelled (e.g. geophysical and atmospherical corrections, long-term time-correlated effects in POD). It contains only uncertainties due to the short-term time-correlated

effects already presented in the Introduction due to differences in the altimeter measurements, in the POD, and in the mean sea surface. To characterise these uncertainties, we calculated the variance and time correlation of GMSL differences during a tandem phase (see Section 3.1). In Fig. 2, the GMSL differences (middle panel) are plotted during the tandem phase of Jason-1 and Jason-2, Jason-2 and Jason-3, and Jason-3 and S6-MF (over the side B period). For the Jason-3 and S6-MF tandem phase, a 2-month periodic signal is observed in the GMSL differences. This signal is currently being investigated and may be attributed

to discrepancies in the precise orbit determination between the two missions. Assuming that this discrepancy will be resolved in the future, we have removed this 2-month periodic signal before calculating the standard deviation of the GMSL differences (dashed line in the middle right panel of Fig. 2). The standard deviations derived from these three tandem phases are close together for Jason-2 and Jason-3, and Jason-3 and S6-MF (0.41 and 0.48 mm, respectively) and slightly higher for Jason-1 and Jason-2 (0.69 mm), most probably due to additional errors in the precise orbit calculation of Jason-1. In the bottom panel

of Fig. 2, the autocorrelation of each GMSL difference does not show an obvious time dependency. We have calculated the number of independent measurements ($n$) of GMSL differences (displayed in the legend of the bottom panel of Fig. 2). The methodology applied to calculate $n$ is the same as in Guérou et al. (2022) to calculate the uncertainty of the GMSL offset during a tandem phase. For the 3 tandem phases, the number of independent measurements is high (e.g. $n = 18$ for a total of 20 measurements (nsample) for S6-MF and Jason3). On the basis of these analyses, we assume that the GMSL differences

are fully decorrelated beyond one month during these three tandem phases analysed. Therefore, we established the uncertainty





budget of the GMSL differences during a tandem phase by a unique 1-month time-correlated error with a standard deviation of 0.5 mm (see Tab.1).

### 3.3 Uncertainty budget at regional scales

The uncertainty budget of sea level differences provided in Section 3.2 on global scale can also be established on regional
spatial scales from a few hundred to a few thousand kilometres. It is deduced from the regional sea level budget uncertainty budget of (Prandi et al., 2021). As on the global scale, most of the different sources of uncertainties described for the regional sea level uncertainty budget are cancelled for the uncertainty budget of regional sea level differences during a tandem phase. Therefore, the uncertainty budget of regional sea level differences during a tandem phase contains only uncertainties due to short-term time-correlated errors. We calculated regional sea level differences between Jason-3 and S6-MF during the tandem
phase for different cell sizes varying from 3° by 3° to 36° by 36° (i.e., from about 300 km by 300 km to approximately 4000 km by 4000 km). Then the standard deviation of the sea level differences is calculated in each cell of varying size. The standard deviation is assigned to the uncertainty budget for the 1-month correlated error (see Tab. 2), homogeneously to the global scale.

## 4 Uncertainty of the 2-tandem phase method

### 4.1 On the global scale

The uncertainty of the trend in sea level differences of the 2-tandem phase method is calculated on the global scale with the uncertainty budget described in Tab. 1. The duration of the first tandem phase is set at 6 months, corresponding to the duration of the tandem phase between Jason-3 and the altimeter side-B of S6-MF (September 14, 2021 to April 07, 2022). For the second tandem phase, we analyse the impact on the trend uncertainty of both the duration of the second tandem phase and the time elapsed since the first tandem phase. To construct the error covariance matrix ($\Sigma$, see Eq. 3), we use the same uncertainty
budget for both tandem phases (see Tab. 1). Fig. 3 shows the evolution of the uncertainty of the trend in GMSL differences as a function of the time period between the two tandem phases (between 1 and 6 years), for 4 different time spans of the second tandem phase (from 1 month to 6 months). For a 1-year time span between the two tandem phases, the uncertainties range from 0.27 mm yr$^{-1}$ for a second phase duration of 6 months to 0.41 mm yr$^{-1}$ for a second phase duration of 1 month. They are reduced to about 0.12-0.16 mm yr$^{-1}$ after 3 years between the two tandem phases. They are less than 0.10 mm yr$^{-1}$ after
5 years between the two tandem phases with a interval of 0.02 mm yr$^{-1}$. These results show that the uncertainty of the trend in the GMSL differences is less influenced by the duration of the second tandem phase when the time elapsed between the two tandem phases increases. For instance, for a time span of 2 years, the uncertainties in the trend are ranging from 0.24 to 10.16 mm yr$^{-1}$ for a second tandem phase respectively from 1 month to 6 months. Whereas for a time span of 5 years, the uncertainties in the trend are ranging from 0.10 to 0.08 mm yr$^{-1}$. Following this analysis, a second tandem phase of 4 months
is deemed sufficient to verify the instrumental stability on the global scale. It should be noted that similar analyses carried out before the launch of S6-MF had already led to this recommendation (Ablain et al., 2020), helping space agencies specify the





scenario for the second tandem phase between Jason-3 and S6-MF. The scenario selected by space agencies (4 months over January-April 2025, Ferrier (2023)) is shown with a star in Fig. 3. Finally, the uncertainty of the trend in GMSL differences with the adopted scenario is 0.14 mm yr$^{-1}$. By construction, the results obtained depend on the specification of the uncertainty

budget (Tab. 1). Therefore, we performed a sensitivity test for the 2-tandem phase method, by varying the temporal correlation of errors between 0 and 2 months instead of 1 month as specified in Tab. 1), while maintaining the same level of variance. In this case, and for the scenario adopted between S6-MF and Jason-3 for the second tandem phase, the uncertainties on the trend vary from 0.06 to 0.18 mm yr$^{-1}$. We also vary the variance from 0.4 to 0.6 mm with a temporal correlation of 1 month. In this case, the uncertainties on the trend vary from 0.11 to 0.17 mm yr$^{-1}$. In the worst case, corresponding to a temporal correlation

of 2 months and a variance of 0.6 mm, the uncertainty on the trend reaches 0.21 mm yr$^{-1}$. Thus, this analysis shows fairly low sensitivity and consolidates the results obtained.

## 4.2 At regional scales

The uncertainty of the trend in sea level differences of the 2-tandem phase method is calculated at regional scales with the uncertainty budget described in Tab. 2. The duration of the first tandem phase remains fixed to 6 months. We also fixed the

210 duration of the second tandem phase to 4 months. We also use the same uncertainty budget for both tandem phases specified at regional scales (see Tab. 2) to construct the error covariance matrix ($\Sigma$, see Eq. 3). Different spatial scales are analysed with cell sizes ranging from 3° x 3° ( 330 km x 330 km) to 36° x 36° ( 4000 km x 4000 km corresponding to regional scales). Fig. 4 shows the evolution of the uncertainties of the trend in regional mean sea level differences for these different configurations as a function of the time elapsed between the two tandem phases. Uncertainties decrease with increasing cell size. After 2 years

and 9 months, corresponding to the scenario adopted for the second tandem phase between S6-MF and Jason-3, the uncertainty ranges from 1.1 mm for a size box of 3° x 3° ( 330 km x 330 km) to 0.4 mm for a size box 36° x 36° ( 4000 km x 4000 km). Detecting trends in regional mean sea level differences less than 1 mm yr$^{-1}$ is possible in almost all ocean basins with a second tandem phase as early as 2 years after the first.

## 5 Comparison with other validation methods

## 5.1 Uncertainty budgets of other validation methods

In the results section, we compare the uncertainty of the 2-tandem phase method with two other existing methods. They are based on sea level comparisons between two altimetry missions on different orbits outside a tandem phase (so-called without-tandem phase method hereafter) and on sea level comparisons between altimeter and tide gauge data (Valladeau et al., 2012; Watson et al., 2015). We have selected these two validation methods because the ability of these two approaches to detect drift

in GMSL is documented in the literature using a similar approach as in this study, based on the establishment of an uncertainty budget.





For the without-tandem phase method, the uncertainty budget has been provided by Jugier et al. (2022) on the global scale and at regional scales between Jason-3 and Sentinel-3 A and B. As the altimeter measurements are not performed exactly at the same time and location, several effects are not cancelled in sea level differences (e.g. geophysical and atmospheric effects, oceanic variability effects, and long-period time-correlated errors in the POD). They contribute to additional sources of uncertainty in the uncertainty budget to assess instrumental stability. We applied the same uncertainty budgets in this study, provided on the global scale in Tab. B1 and at regional scales in Tab. C1.

For the method based on sea level comparisons between altimeter and tide gauge data, the uncertainty budget was provided by Ablain et al. (2018). As for the without-tandem phase method, the altimeter and tide gauge measurements are not performed exactly at the same time and location. Furthermore, all the errors performed by two independent system measurements are not cancelled in the sea level differences. Therefore, the uncertainty budget contains an additional source of uncertainties related to the geophysical and atmospheric effects, oceanic variability effects, long-period time-correlated errors (> 5 years) in the altimeter measurements (e.g., POD) and the tide gauge measurements (e.g., land motion correction). They also contribute to additional sources of uncertainty in the uncertainty budget to assess instrumental stability. The uncertainty budget is provided at the global scale in Tab. A1. On regional scales, the method based on the comparison of altimetry and tide gauge data is generally not suitable for detecting precise regional sea level drifts. This is mainly due to uncertainties associated with the vertical ground motion of the tide gauge records and the distance of the tide gauges from the nearest altimetry measurement (Valladeau et al., 2012; Watson et al., 2021). It should be noted that for some very specific locations, the tide gauges are very well referenced, allowing for more accurate local comparisons (e.g. Mertikas et al. (2021)).

## 5.2 On the global scale

We evaluated the ability of the 2-tandem phase method to detect a trend in the GMSL differences by comparing the uncertainties obtained with the two other validation methods: a) sea level comparisons between two altimetry missions on different orbits without tandem phase, and b) sea level comparisons between altimeter and tide gauge data. In Fig. 5, the uncertainties have been plotted for the three methods as a function of the time spent between the two tandem phases of Jason-3 and S6-MF (2 years and 9 months). The duration of the second tandem phase has been set at four months, according to the adopted scenario Ferrier (2023). It is also worth noting that the total duration of the time series used to calculate the uncertainties for all methods includes the duration of both tandem phases. This means that for 1 year spent between the two tandem phases, the total length of the time series is 1 year and 10 months (1 year + 6 months + 4 months). Similarly, for the adopted scenario of a second tandem phase of 4 months, 2 years, and 9 months after the first of 6 months, the total duration of the analysed time series is 3 years and 7 months. The analysis in Fig. 5 clearly shows that the 2-tandem phase method significantly reduces the uncertainties of the trend in the GMSL differences. Over a period of 3 years and 7 months, which corresponds to the adopted scenario, the uncertainty is 1.2 mm yr$^{-1}$ with the tide gauge method. It is reduced to 0.42 mm yr$^{-1}$ for the without tandem phase method, whereas with the new 2-tandem phase method, the uncertainty decreases to 0.14 mm yr$^{-1}$. Increasing the time spent between the two Jason-3/S6-MF tandem phases to 4 years, and hence the total duration to 4 years and 10 months, leads to lower



uncertainties for each method at 0.9 mm yr$^{-1}$, 0.35 mm yr$^{-1}$ and 0.1 mm yr$^{-1}$ respectively. These results highlight the better ability of the two-tandem-phase method to assess instrumental stability on the global scale.

### 5.3   At regional scales

Similarly to the global scale, we compare the 2-tandem phase method with the method based on sea level comparisons between two altimetry missions on different orbits (outside a tandem phase). In Fig. 6 we evaluate the evolution of the uncertainty of

the trend in the regional mean sea level differences of the two methods with the time elapsed between two tandem phases, after selecting size boxes of 9° x 9° (about 1000 km). As uncertainties are not spatially homogeneous (see Fig. A1), we have also plotted in Fig. 6 the envelope of the spatial distribution of uncertainties at values between the 16th and 84th percentile (i.e., 1-$\sigma$). For a cell size of 9° by 9° (about 1000 km by 1000 km), the uncertainty at 1-$\sigma$ ranges from 1.8 to 4.8 mm, with a median of 2.3 mm. We observe that the average value of trend uncertainty is even more significantly reduced at regional scales than

on the global scale with the 2-tandem phase method. By conducting the analysis over 3 years and 7 months, corresponding to the adopted scenario of a second tandem phase 2 years and 9 months after the initial one, the trend uncertainty of sea level differences is 4.0 mm yr$^{-1}$ without tandem phase, while with the new 2-tandem phase method the uncertainty decreases to 0.65 mm yr$^{-1}$. Increasing the total duration of the analysis to 4 years and 10 months (corresponding to 4 years between the two tandem phases of Jason-3 and S6-MF) results in lower uncertainties for each validation method, 3.0 and 0.5 mm yr$^{-1}$,

respectively. Significantly better results obtained at regional scales are mainly explained by the effect of ocean variability. As mentioned above, it is a significant source of uncertainty when measurements are not collocated in time and space. At regional scales, the effect of ocean variability is not spatially averaged and becomes a major source of uncertainty. As the 2-tandem phase method is not affected by this effect, its ability to assess instrumental stability on regional scales is very promising.

### 6   Conclusions

We have proposed a novel cross-validation method to assess the instrumental stability, based on the realisation of a second tandem phase between 2 successive reference missions a few years after the initial one. We have demonstrated the ability of the 2-tandem phase method to assess the instrumental stability. Assuming a second tandem phase with a minimal duration of 4 months, it will be possible to assess on the global scale the instrumental stability with an uncertainty of less than ±0.1 mm yr$^{-1}$ in a CL of [16-84]% for time periods between the two tandem phases of 4 years and beyond. This means 3 to 8

285   times better than with the other validation method based, respectively, on sea level comparisons outside a tandem phase and on altimetry and tide gauge comparisons. On regional scales, the gain is greater with an uncertainty of ±0.5 mm yr$^{-1}$ in a CL of [16-84]% for spatial scales of about 1000 km, that is, 6 times better than with the method based on sea level comparisons outside a tandem phase. The 2-tandem phase method could be applied for the first time between S6-MF and Jason-3 after the realisation of the second tandem phase early in 2025. On the global scale and for the scenario adopted for the second tandem

phase between Jason-3 and S6-MF, it will be possible to assess sea level stability with an uncertainty of ±0.14 mm yr$^{-1}$ in a CL of [16-84]%. On regional scales, it will be possible to assess sea level stability with an uncertainty of ±0.65 mm yr$^{-1}$



in a CL of [16-84]% for spatial scales of about 1000 km. To date, we have assumed that the uncertainty budget of relative errors will be the same over the two tandem phases between S6-MF and J3. It should be re-evaluated after the completion of the second tandem phase, and the uncertainties in the trend may be updated. If a significant instrumental drift in sea level is 295 detected, it cannot automatically be attributed to the S6-MF or Jason-3 altimetry missions. In-depth investigations will have to be carried out by experts from the two satellites, and alternative validation methods will also have to be implemented, even if they are less accurate at detecting a drift in altimeter measurements. These inquiries could lead to revisions in the mean sea level uncertainty budget to reflect uncertainties in the instrumental stability.

The performances of the 2-tandem phase method are very promising, however, the novel method also presents some limita-300 tions. The method is only applicable over the period encompassing the 2 tandem phase and does not allow the assessment of the altimeter stability outside this period. This method only allows the assessment of the stability of instrumental errors since since all the other sources errors have been cancelled (e.g., geophysical and atmospherical effect, long-term errors in the POD). Therefore, no assessment of other sources of errors is made in the uncertainty budget of the mean sea level (Ablain et al., 2019a; Guérou et al., 2023; Prandi et al., 2021). For these reasons, the other validation methods are complementary, although 305 their uncertainties in the trend of sea level differences are higher.

In order to take a larger benefit of this novel cross-validation method, this involves regularly implementing double tandem phases between two successive altimetry missions in the future. The instrumental stability could be assessed over a long period. Assuming that it is theoretically possible to organise a third tandem phase between Jason-3 and S6-MF six years after the initial tandem phase, we could assess instrumental stability with uncertainties of $\pm 0.07$ mm yr$^{-1}$ ([16-84]% CL) on the global scale 310 (see Fig. 5), and approximately $\pm 0.3$ mm yr$^{-1}$ ([16-84]% CL) at regional scales (see Fig. 4). The feasibility of conducting regular tandem phases between reference missions needs to be analysed considering the end-of-life limitations of the reference missions.

*Author contributions.* M.A and G.D. conceived the presented approach. M.A., N.L., R.F. developed the theory and performed the computations. All authors discussed the results and contributed to the final manuscript.

*Competing interests.* None declared

*Acknowledgements.* This study was initiated as part of the SALP project supported by CNES, the preliminary results of which were presented by (Ablain et al., 2020) and helped the space agencies define the scenario selected for the second tandem phase between Jason-3 and S6-MF. The study was then completed and updated as part of the S6JTEX project supported by ESA. It is also important to mention that the ASELSU project supported by ESA contributed to improving the formalism used in this article to provide uncertainties.



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





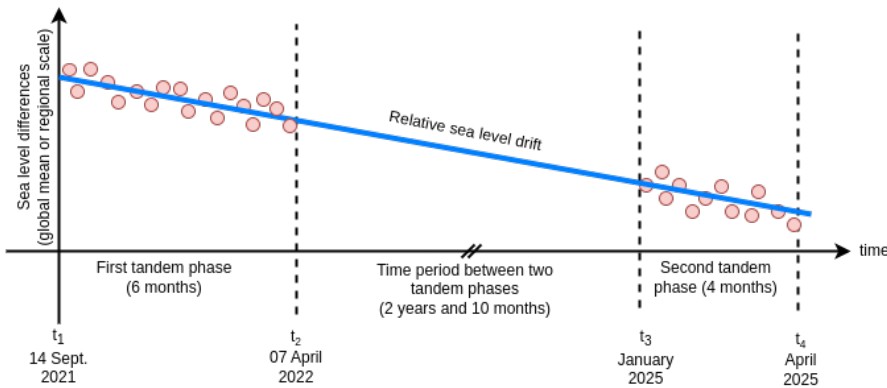

**Figure 1.** Basic principle of the 2-tandem phase method applied to the Jason-3 and S6-MF altimetry satellites.

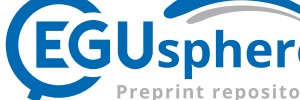

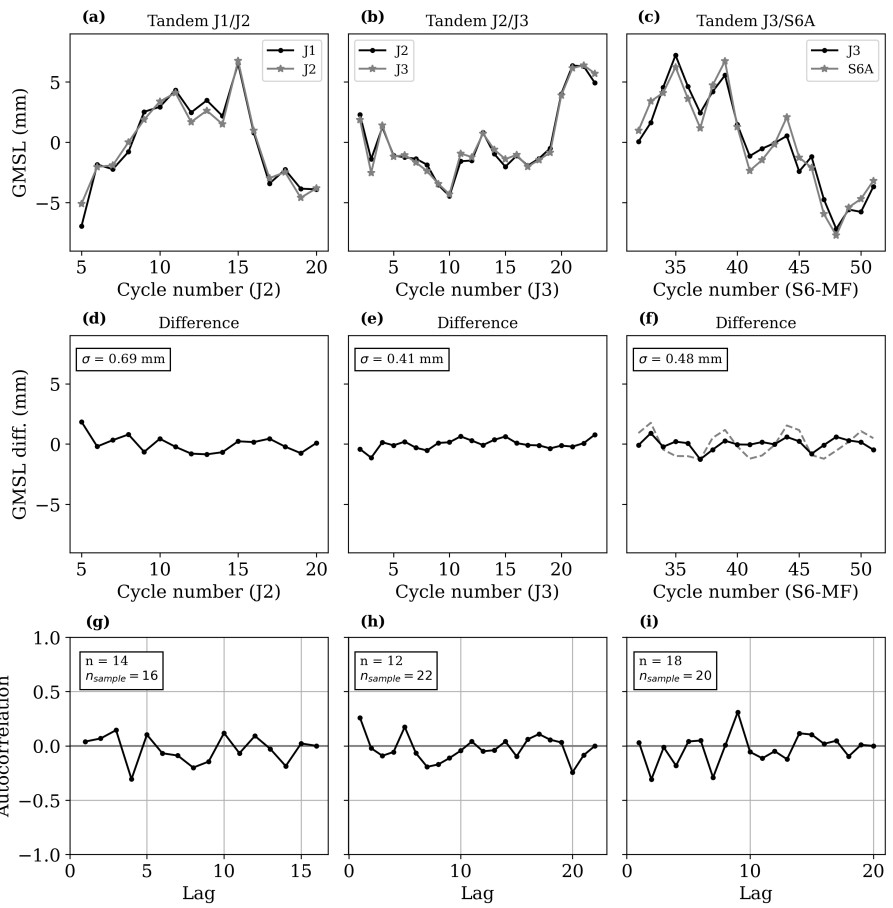

**Figure 2.** Panels (a-c) show GMSL records over the tandem phase between Jason-1 and Jason-2 on left (noted J1/J2), between Jason-2 and Jason-3 on the centre (noted J2/J3) , between Jason-3 and S6-MF (side B) on the right (noted J3/S6-MF); (d-f) show the GMSL record differences between the two respective missions in tandem phase. $\sigma$ is the standard deviation of the GMSL differences; and (g-i) show the autocorrelation of the difference signal with n the number of independent measurements of the sample and $n_{sample}$ the total number of measurements of the sample. The mean value of each time series has been removed to facilitate comparison.





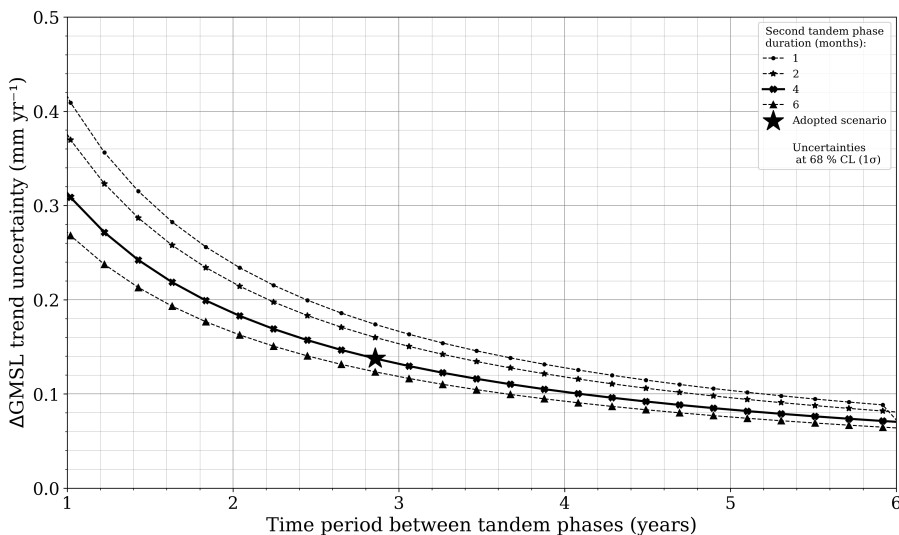

**Figure 3.** Evolution of the uncertainty of the trend in GMSL differences (ΔGMSL) with the time elapsed between the two tandem phases between Jason-3 and S6-MF for several durations of the second tandem phase, ranging from 1 month to 6 months. The scenario adopted by the spatial agencies for the second tandem phase, which is four months long and separated by two years and nine months from the first, is indicated with a star.





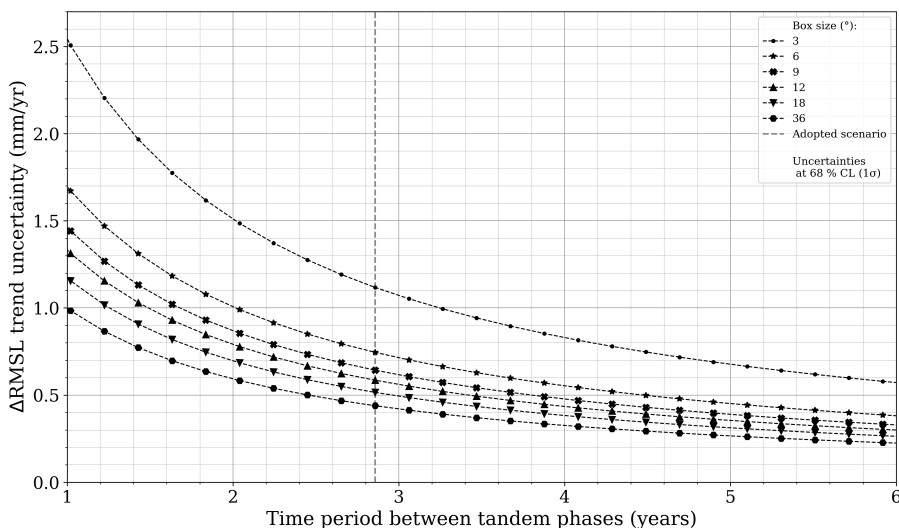

**Figure 4.** Evolution of the uncertainty of the trend in regional mean sea level differences (ΔRMSL) with the time period between the two tandem phases between Jason-3 and S6-MF for different cell sizes from 3°x3° (corresponding to 330 km spatial scale) to 36°x36° (corresponding to 4000 km spatial scales).



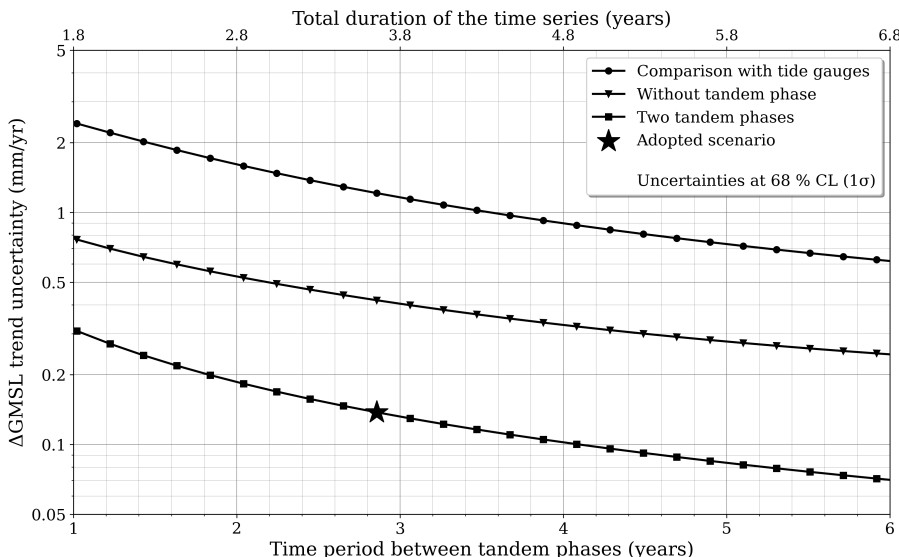

**Figure 5.** Evolution of the uncertainty of the trend in GMSL differences ($\Delta$GMSL) with the time spent between the two tandem phases of Jason-3 and S6-MF for the different calibration methods: a) comparison with tide gauges ; b) inter-mission comparison without tandem; c) two tandem phases. The scenario adopted for the second tandem phase, which is four months long and separated by two years from the first one, is indicated with a star. The y-axis scale is logarithmic.



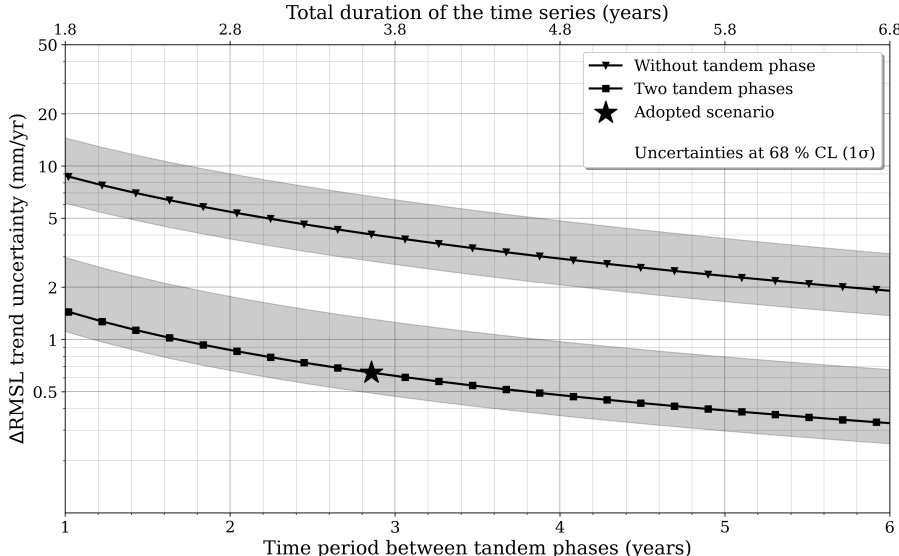

**Figure 6.** Evolution of the trend uncertainty of regional mean sea level differences (ΔRMSL) differences with the time period between the two tandem phases between Jason-3 and S6-MF for different calibration methods and cell size of 9°x9° (corresponding to 1000 km spatial scale). The envelope represents the spatial distribution of uncertainties between the 16th and 84th percentile (i.e., 1-$\sigma$ ) values. The y-axis scale is logarithmic. The scenario adopted for the second tandem phase, which is four months long and separated by two years and 9 months from the first one, is indicated with a star.





**Table 1.** Uncertainty budget of GMSL differences between two altimetry missions in tandem.

| Source of uncertainty | | Time correlation of errors | Uncertainties (1-$\sigma$) |
|---|---|---|---|
| short-term time-correlated errors due to altimeter processing, precise orbit determination, etc. | | short-term time-correlated errors $\lambda <$ 1 month | $U_\sigma$ = 0.7 mm for Jason-1/Jason-2 $U_\sigma$ = 0.4 mm for Jason-2/Jason-3 $U_\sigma$ = 0.5 mm for Jason-3/S6-MF[1] |
| | | short-term time-correlated errors 1 month < $\lambda <$ 1 year | $U_\sigma$ = 0: no uncertainty identified |
| Stability of the wet tropospheric correction (WTC) | | long-term time-correlated errors $\lambda <$ 5 years | $U_\sigma$ = 0: model WTC are used to cancel WTC errors in GMSL differences |
| Precise orbit determination stability | International Terrestrial Reference System (ITRF) | Linear time-correlated errors | $U_\delta$ = 0: errors are cancelled between two missions in tandem |
| | Gravity fields | Long-term time-correlated errors $\lambda <$ 10 years | $U_\sigma$ = 0: errors are cancelled between two missions in tandem |
| GIA correction | | Linear time-correlated errors | $U_\delta$ = 0: errors are cancelled between two missions in tandem |

[1] The uncertainty budget in this study is constructed by taking the U$\sigma$ for Jason-3/S6-MF



**Table 2.** Uncertainty budget of regional sea level differences between two altimetry missions in tandem. Values are provided for 9° × 9° box sizes within a [16-84]% confidence level.

| Source of uncertainty | Time correlation of errors | Uncertainty (1-$\sigma$) |
|---|---|---|
| Short-term time-correlated errors due to altimeter processing, precise orbit determination, oceanic variability, etc .. | short-term time-correlated errors $\lambda$ <1 month | $U_\sigma \in [1.8, 4.8]^1$ mm Location dependent |
| Stability of the wet tropospheric correction (WTC) | Long-term time-correlated errors $\lambda$ <5 years | $U_\sigma = 0$: model WTC are used to cancel WTC errors in sea level differences |
| Precise orbit determination stability | Linear time-correlated errors | $U_\delta = 0$ |
| Instrumental stability | Linear time-correlated errors | $U_\delta = 0$ |
| GIA correction | Linear time-correlated errors | $U_\delta = 0$ |

[1] The uncertainty budget in this study is constructed by taking the median value : $U_\sigma$ = 2.3 mm for $\lambda$ < 1 month.



**Table A1.** Uncertainty budget of GMSL differences between altimeter measurements and tide gauges data from GLOSS/CLIVAR network (from Ablain et al. (2018)).

| Source of uncertainty | Time correlation of errors | Uncertainty (1-$\sigma$) |
|---|---|---|
| Short-term time-correlated errors due to tide gauge and altimeter measurement errors, but also due to the collocation of both datasets. | Short-term time-correlated errors $\lambda < 6$ months | $U_\sigma$ = 4.0 mm for T/P<br>$U_\sigma$ = 3.5 mm for Jason-1<br>$U_\sigma$ = 2.3 mm for Jason-2/Jason-3[1] |
| | Short-term time-correlated errors 6 months $< \lambda < 1$ year | $U_\sigma$ = 1.0 mm for T/P<br>$U_\sigma$ = 0.7 mm for Jason-1<br>$U_\sigma$ = 0.5 mm for Jason-2/Jason-3[1] |
| Long-term time correlated errors due to tide gauge networks (e.g. averaging method to take into spatial distribution), long-term stability of tide gauge time serie | Long-term time-correlated errors $\lambda < 3$ years | $U_\delta$ = 0.1*$\sqrt{2}$ mm yr$^{-1}$ |
| | Long-term time-correlated errors $\lambda < 10$ years | $U_\sigma$ = 1.0 mm |
| Linear time-correlated over all the altimetry period due to the VLM errors of the tide-gauge network. | Linear time-correlated errors | $U_\delta$ = 0.2 mm yr$^{-1}$ |

[1] The uncertainty budget in this study is constructed by taking the $U_\sigma$ for Jason-2/Jason-3





**Table B1.** Uncertainty budget of the GMSL differences between two altimetry missions not in tandem (from Jugier et al., 2022).

| Source of uncertainty | | Time correlation of errors | Uncertainties (1-$\sigma$) |
|---|---|---|---|
| Short-term time-correlated errors due to altimeter processing, precise orbit determination, etc. | | Short-term time-correlated errors $\lambda < 2$ months | $U_\sigma \in [0.6, 0.8]^1$ mm Depending on altimetry missions |
| | | Short-term time-correlated errors 2 months $< \lambda < 1$ year | $U_\sigma \in [0.5, 0.7]^1$ mm Depending on altimetry missions |
| Stability of the wet tropospheric correction (WTC) | | Long-term time-correlated errors $\lambda < 5$ years | $U_\sigma = 0$: model WTC are used to cancel WTC errors in GMSL differences |
| Precise orbit determination stability | International Terrestrial Reference System (ITRF) | Linear time-correlated errors | $U_\delta = 0.1 * \sqrt{2}$ mm yr$^{-1}$ |
| | Gravity fields | Long-term time-correlated errors $\lambda < 10$ years | $U_\sigma = 0.5 * \sqrt{2}$ mm yr$^{-1}$ |
| GIA correction | | Linear time-correlated errors | $U_\delta = 0$ |

[1] The uncertainty budget in this study is constructed by taking the mean value of the range : $U_\sigma = 0.7$ mm for $\lambda < 2$ months and $U_\sigma = 0.6$ mm for 2 months $< \lambda <$ 1 year.



**Table C1.** Uncertainty budget of the MSL differences between two altimetry missions not in tandem (from Jugier et al., 2022). Values are provided for 9° × 9° box sizes within a 16th-percentile and 84th-percentile interval.

| Source of uncertainty | Time correlation of errors | Uncertainties (1-$\sigma$) |
|---|---|---|
| Short-term time-correlated errors due to altimeter processing, precise orbit determination, etc. | Short-term time-correlated errors $\lambda < 2$ months | $U_\sigma \in [0.6, 0.8]^1$ mm Depending on altimetry missions |
| | short-term time-correlated errors 2 months $< \lambda < 1$ year | $U_\sigma \in [0.5, 0.7]^1$ mm Depending on altimetry missions |
| Stability of the wet tropospheric correction (WTC) | Long-term time-correlated errors $\lambda < 5$ years | $U_\sigma = 0$: model WTC are used to cancel WTC errors in GMSL differences |
| Precise orbit determination stability | Linear time-correlated errors | $U_\delta = 0.33*\sqrt{2}$ mm yr$^{-1}$ |
| Gravity fields | Long-term time-correlated errors $\lambda < 10$ years | $U_\sigma = 0.5*\sqrt{2}$ mm yr$^{-1}$ |
| GIA correction | Linear time-correlated errors | $U_\delta = 0$ |

[1] The uncertainty budget in this study is constructed by taking the median value : $U_\sigma = 9.4$ mm for $\lambda < 2$ months and $U_\sigma = 4.9$ mm for 2 months $< \lambda < 1$ year.



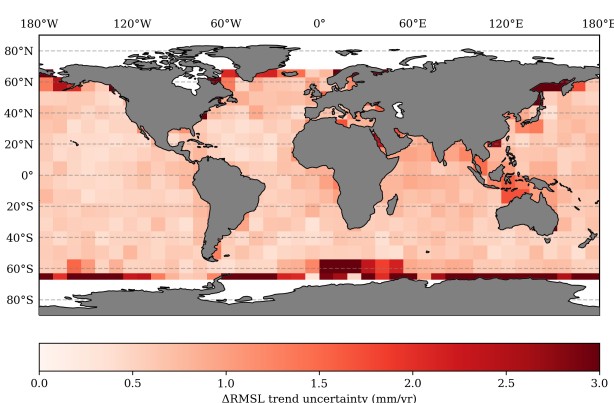

**Figure A1.** Uncertainties of the trend in regional mean sea level differences of the 2-tandem phase method for a 9° x 9° cell size and for a 2-year and 9 month time spent between the two Jason-3 and S6-MF tandem phases.