# Peer review of "Benefits of a second tandem flight phase between two successive satellite altimetry missions for assessing the instrumental stability"

_EGUsphere, 2024_

## Author Comment (AC1)

**Reviewer #2 ([https://doi.org/10.5194/egusphere-2024-1802-RC2](https://doi.org/10.5194/egusphere-2024-1802-RC2))**

Review of "Benefits of a second tandem flight phase between two successive satellite altimetry missions for assessing the instrumental stability" by Ablain et al.

This papers describes a potential second tandem flight phase and the benefits this would bring to the calibration of two reference altimeters.

While the topic is of interest, the paper is often too brief in describing what has been done, and the reader needs to look for the cited references in order to understand the methodology (and sometimes even those references don't give the information we're after). For example, in section "5.1 Uncertainty budgets of other validation methods" the authors say "the altimeter measurements are not performed exactly at the same time and location". There is no information about the maximum allowed differences in space and time between altimeter and tide gauge data, or between 2 altimeters, so the reader is unable to assess how this comparison is done. The cited references do not give this information either, or I have been unable to find them (one of the references, Ablain et al 2018, is not complete in the list of references and I have been unable to find it).

**For the sake of clarity, we have merged Sections 2 and 3, which provide a sequential explanation of each step in the proposed methodology for calculating the uncertainty of the 2-tandem phase method. This restructuring is intended to facilitate a better understanding of the methodology.**

**Although not the main focus of this study, the methodology for deriving the uncertainty of comparison methods between altimetry and tide gauges, or between 2 altimetry missions outside a tandem phase, is also based on the same methodology as the 2-tandem phase method, with the provision of a specific uncertainty budget. By clarifying the methodology for the 2-tandem phase method (new section 2), we also aim to shed light on the other comparison methods. We have also clarified the text and mentioned the typical time differences and distance differences.**

**We have also completed the reference "Ablain et al., 2018" has been updated provdiding the access the to the file: [https://www.geomatlab.tuc.gr/fileadmin/users_data/geomatlab/international_review_workshop_2018/presentations/01_Monday/Session_01/06_S1_23_Ablain_et_al.pdf](https://www.geomatlab.tuc.gr/fileadmin/users_data/geomatlab/international_review_workshop_2018/presentations/01_Monday/Session_01/06_S1_23_Ablain_et_al.pdf)**

The manuscript also contains various vague statements or definitions, in particular when using the word "uncertainty" throughout the manuscript. It should be clear at

each moment of the paper which uncertainty are they talking about (example, the title of section 2 "Method to quantify the uncertainty of the 2-tandem phase method", which uncertainty? As it is phrased, it sounds like the uncertainty that the 2-tandem phase takes place or not...). Also, there are  2 "method" in the same sentence. The term "estimated" should also go with uncertainty in many parts of the manuscript.

**Corrected: The title of this section has been updated to "Methodology for estimating uncertainty in the 2-tandem phase method" to clearly indicate that it refers to the process of estimating the uncertainty associated with this method, not uncertainty regarding the occurrence of the second tandem phase itself. Additionally, the introduction of this section has been revised to further clarify this distinction.**

**The use of the term "uncertainty" throughout the manuscript has been carefully reviewed. We ensure that throughout the manuscript, the types of uncertainties and their associated sources (e.g., instrumental noise, geophysical corrections, etc.) are consistently defined and referenced.**

I'll specify my comments in order of appearance:

Line 39, but general comment: please check the parentheses of your citations, they are often wrong ("identified and characterised by (Ablain et al., 2019b; Guérou et al., 2023; Prandi et al., 2021)." should be "identified and characterised by Ablain et al. (2019b), Guérou et al. (2023) and Prandi et al. (2021)."

**We have corrected in the paper the parentheses of our citations.**

line 81: "drifts in the seal level data record". Strange wording, it is not the sea level that drifts (it changes, and it is not that change the authors are after) but the satellite sensors' accuracies.

**Indeed, the sentence was wrong, we are looking for drift errors.  This has been corrected.**

Line 116: "This uncertainty budget enumerates the various sources of uncertainty", please enumerate which ones. Line 117: "standard deviation associated with errors": which errors are you talking about here?

**The main sources of errors in the stability of the sea level data record have been detailed in the introduction. They are attributed to short-term time-correlated errors, long-term time-correlated errors and offset between two successive reference missions. The sentence has been clarified in the updated document.**

Line 120 \Sigma_TP should be \Sigma_tp

**Corrected**

Equations (and text), please put matrices in bold characters, especially \Sigma since it is used later as the summation symbol.

**Corrected**

Line 136 The equation there is given without describing each of its components, and what the components i of the correction are.

**We have detailed each component of the equation and added a description.**

Line 128: "relative errors observed during the two tandem phases". No information is given about these "relative errors", which errors and how are they estimated/calculated. Also, in line 134 it says "we can study the uncertainty of the 2-tandem phase method without having yet executed the second tandem phase" so the use of "observed" in the first sentence is a bit misleading.

- **We have merged sections 2 and 3, presenting a sequential explanation of each step of the methodology proposed to calculate the uncertainty of the 2-tandem phase method. This reorganization aims to facilitate a clearer understanding of the methodology.**
- **In section 2.2, we have added detailed information on the calculation of error covariance matrices based on the work of Ablain et al., 2019. We have clarified the construction of the covariance matrix during the second tandem phase**

Line 142: the choice of 1 degree in latitude and 3 degree in longitude has been used elsewhere, but some explanation about this choice should be provided.

**The 1° latitude per 3° longitude choice has been made following recommendation by Henry et al. (2014) to optimise the effect of sea level variability observed by reference altimeter missions in the GMSL. A sentence to clarify this has been added to the manuscript.**

Line 145. Explain the GMSL AVISO method briefly

**The AVISO method have been detailed and added to Section 2.1.1**

Line 154 (and figure 2). A 2-month periodic signal is mentioned in panel f. The attribution to POD is given without reasoning, and in fact POD uncertainties are part of the uncertainty budget you are trying to assess, so why removing it?

**During the tandem phase, we observed a periodic signal in the GMSL differences between J3 and S6-MF. This signal, occurring over a two-month period, is attributed to differences in the Precise Orbit Determination (POD) calculations used for each satellite.**

**Recent research by Cadier et al. (submitted) has identified the source of this periodic signal as β′ (beta-prime) dependencies within the CNES POD solution employed for S6MF. Their work demonstrates that using the JPL POD solution significantly reduces this 60-day error signal (see figure below from Cadier et al.).**

**While systematic GMSL differences arising from POD variations can occur during tandem phases, these discrepancies are typically resolved a few years after the tandem phase ends. This correction is achieved through adjustments to the POD calculations by POD experts, as has been the case for all tandem phases involving Topex/Poseidon, Jason-1, Jason-2, and Jason-3. Therefore, we consider the uncertainty due to POD discrepancies to be negligible in this study.**

**The paper offers now a more comprehensive justification for the aforementioned observation regarding the POD differences and the contribution of POD uncertainty.**

[Figure]

Fig. 26. GMSL differences between J3 and S6-MF LR NR over the side B tandem phase using the radiometer and CNES orbits (a), model and CNES orbits (b), model and JPL orbits (c)

Also, you say that the data without this 2-month signal is the dashed line in figure 2f, but for me this line shows a much more evident periodic signal. Further down the text it is mentioned that "the autocorrelation of each GMSL difference does not show an **obvious** time dependency", and that the "the GMSL differences are fully decorrelated beyond one month" which is expected since I guess this has been done after removing this 2-month signal?

**The dashed line is the difference before removing the periodic signal. It has been clarified in the text and added on the figure caption. The autocorrelation plot has been done after removing the signal for Jason-3 and Sentinel-6A.**

Line 176 "The standard deviation is assigned to the uncertainty budget for the 1-month correlated error (see Tab. 2), homogeneously to the global scale." Sentence not complete?

**Corrected**

line 193: 10.16 mm yr-1 sounds huge

**Corrected**

Line 229: provide details about the uncertainty budgets and what are the maximum delta_t and delta_{x,y} used to compare 2 altimeters and an altimeter to tide gauge data

**We do not think it is the aim of this paper to provide too much detail on the uncertainty budget of other comparison methods (we have already provided the uncertainty table). However, the text has been significantly improved and we also provide the typical figures for the time difference and spatial difference.**

Line 246 You repeat that you are doing two additional methods (line 246-248)

**Corrected**

line 324 Incomplete reference (pages, journal?)

**Corrected : we have provided the accessthe file: https://www.geomatlab.tuc.gr/fileadmin/users_data/geomatlab/international _review_workshop_2018/presentations/01_Monday/Session_01/06_S1_23_Ablain _et_al.pdf**

Figure 1. Line blue is the "relative sea level drift" Again, I think this wording is confusing.

**Corrected: it has been replaced by "Relative sea level differences drift"**

Figure 2. central figures should have a reduced y-axis to see better the variability. Same for bottom panels.

**Corrected**

Indicate what the dashed line is in 2f.

**Corrected**

Figures 3, 4 and 5: the symbols are quite small and difficult to tell apart (especially in figures 3 and 4). The legend contains "uncertainty at 68%" but it is not shown in the figures. The vertical and horizontal lines within the figures seem to be added rather randomly, and in figure 4 there is an extra vertical dashed line that is not explained

**Corrected**

The symbols have been enlarged. The "uncertainty at 68%" is to give information on the uncertainty plotted in the figure. The vertical and horizontal lines are the grid of the plot, they have been removed for clarity in figure 3 and 4. The extra vertical dashed line is the adopted scenario for the second tandem phase and is mentioned in the legend. The colour is set to blue to better distinguish it.

---

## Author Comment (AC2)

**Reviewer #1 (https://doi.org/10.5194/egusphere-2024-1802-RC1)**

In this paper the authors present a novel validation method to assess the instrumental drifts in the global mean sea level trends based on the implementation of a second phase in which the two successive reference altimetry satellites fly in tandem a few years after the first. They quantify the uncertainties of the trend such method would achieve depending on the duration of the second tandem flight phase and the length of time between the two phases and compare the results with other validation methods, proving the usefulness of implementing the second tandem phase.

The manuscript represents a significant scientific contribution to achieving stable and consistent sea level measurements. However, while the methods applied in this study are most likely valid, it is hard to fully assess that because they are not presented in a clear way or sometimes at all, leaving some questions about them unanswered.

Major comments

1. It is hard to understand the methodology and follow the paper the way it is written because many parts are not explained, but only reference other papers where the method is described. It is perhaps not necessary to go into every detail as in the papers where the method is used first, but the reader should not need to read several other papers to understand what was done in this one. The paper should include a better explanation of how the covariance matrix for one tandem phase is created and how the number of independent observations is calculated, as well as what that number means for the calculation of the uncertainties. An additional problem is the way the manuscript is structured: Sect. 2 describes the method to quantify the uncertainty of the 2-tandem phase, which requires the error covariance matrix for each of the phases. It is not explained well enough how these error covariance matrices are calculated, but also the description of what they contain only comes after, in Sect. 3, which completely disrupts the flow of text.

**In response to the major comments received, we have made the following improvements to our paper:**

- **We have merged sections 2 and 3, presenting a sequential explanation of each step of the methodology proposed to calculate the uncertainty of the 2-tandem phase method. This reorganization aims to facilitate a clearer understanding of the methodology.**
- **In section 2.2, we have added detailed information on the calculation of error covariance matrices based on the work of Ablain et al., 2019.**

- **To enhance clarity, we have provided more detailed explanations of how the number of independent measurements is calculated and how this number is subsequently utilised to derive the uncertainty.**

2. It could be because of my misunderstanding due to lacking methodology explanations, but the sensitivity study and its conclusions do not seem convincing at all. You claim that the sensitivity tests showed that there is fairly low sensitivity of the uncertainty to the temporal correlation of errors and/or the variance because the uncertainty varies between 0.06-0.18 and 0.11-0.17, which correspond to 0.12 and 0.06 range, respectively. However, based on Fig. 2, the uncertainty varies between 0.12 and 0.18 depending on the duration of the second phase (with 2 years and 9 months between phases), so also only 0.06 difference. Same if you change the length between the two tandem phases from 3 to 6 years (double!), the uncertainty is only decreased by 0.06, from 0.13 to 0.07. To me that seems like the uncertainty is a lot more sensitive to the choice of decorrelation time than to the length of the second tandem phase or even the time between the two phases. Or am I misunderstanding something here? It would help to explain better how you chose the 1 month for the temporal correlation to make sure it is a good choice, since it seems to affect the results quite a lot.

**The uncertainty we have estimated does indeed vary as a function of the length of the second tandem phase and the gap between the two tandem phases. One of the aims of this paper is therefore to determine these variations in uncertainty as a function of these two parameters, in order to be able to prescribe a relevant scenario for the second tandem phase between two successive missions. For example, the preliminary results of this study, which will be presented to OSTST in 2020, have been used to determine the minimum duration of the second tandem phase between S6 and J3 (which will take place in the first quarter of 2025).**

**However, the results obtained are sensitive to the uncertainty budget of the SSHA differences during a tandem phase, which we have estimated in this study (Section 3). In order to accurately estimate this uncertainty budget, we have analysed three different tandem phases, from which we have derived the standard deviation of the differences and the temporal correlation of the SSHA differences during a tandem phase (see Section 3). Each tandem phase gives close but not strictly identical results, which can be explained by the differences between the SSHAs of the different altimetry missions compared to each other, and by the relatively short duration of the tandem phases (about 6 to 9 months). Therefore, we thought it would be useful to analyse the sensitivity of our analyses by varying the uncertainty budget by taking the extreme values possible for the uncertainty budget (for example, by varying the temporal correlation from 15 days to 2 months). The aim of this sensitivity analysis is to demonstrate the robustness of the results of this study.**

**In the updated paper we have therefore improved the value of this sensitivity analysis in relation to the uncertainty budget. The fact that the methodological sections have been improved should also clarify this sensitivity analysis.**

Other comments

L61-65: This is a very long and quite hard to read sentence.

**Corrected : The sentence has been split.**

L86: This seems like a wrong unit or a very wrong number.

**Corrected**

L106: dot missing

**Corrected**

L? (p5 bottom): "contain the along-track sea level anomaly at 1Hz (SLA, see Eq. (1))"

What does Eq. (1), which is for the estimator of beta, have to do with SLA? Seems like it is referring to the wrong equation.

**Corrected**

L? (p6 top): "The along-track SLA provided in the L2P products is derived from the following equation:

SLA = Orbit − Range − Σ_i Correction_i − MeanSeaSurface"

What is the meaning of each of the variables in this equation?

**Corrected : The meaning of each variable in the equation has been defined**

L137-145 It is not clear to me whether you here describe how the dataset you downloaded and used was created or the processing steps you applied to the dataset before using it. It is especially unclear what was done here considering the "regional scales" you refer to throughout the rest of the manuscript are using different longitude-latitude box sizes that the 1 degree latitude  per 3 degrees longitude mentioned here. Additional minor comment: You here use degree (word), and in other places in the manuscript degree symbol when describing the size of the box. Please be consistent.

**L2P products  contain the SLA along the track at 1Hz, already homogenised in terms of geophysical and atmospheric correction. On our side, the work**

**consisted in calculating the GMSL differences between two altimeter missions during a tandem phase from the 1Hz SLA measurements along the track available in L2P. We applied the same method as in Henry et al. (2014), who recommend applying 1° latitude per 3° longitude to the calculated SLA grids to optimise the effect of sea level variability observed by reference altimeter missions in the GMSL.**

**In the revised paper, the applied method is better described. We have also used the degree symbol (°) rather than the word "degree".**

L147-152 This is a general statement, not specifically for the global scale, so it should be in the previous subsection.

**Corrected, the sentence has moved in new section 2.1**

L151-152 Why exactly are you referring to the previous section here? Seems completely unnecessary and it disrupts the reading.

**Corrected**

L156-157 1. Is the dashed line the periodic signal or the differences before removing it? The way this sentence is phrased, it could also be understood as the dashed line being the difference with the periodic signal removed, which does not seem the case when looking at the figure. The explanation of what the grey dashed line is should also be in the figure caption; 2. How much did the removal of the signal reduce the standard deviation (or what was the standard deviation before removing this signal)?

**Corrected: the dashed line is the difference before removing the periodic signal. Removing the signal reduces the standard deviation from 0.99 to 0.48 mm. It has been clarified in the text and added on the figure caption.**

L164-167 This could be because there was no proper explanation of what is the meaning of n and its calculation, but I cannot follow this conclusion. Could you please explain the reasoning behind it?

**The temporal signal of GMSL difference between Jason-3 and Sentinel6-MF consists of 20 measurements taken every 10 days. We used the first-lag autocorrelation coefficient, $\rho_1$, to estimate the number of independent measurements using the formula from Guerou et al. (2023):**

$$n = \frac{(1-\rho_1)}{(1+\rho_1)} \times n_{\text{sample}}$$

**where $\rho_1$ is the first-lag autocorrelation coefficient, which represents the correlation between consecutive measurements, and $n_{sample}$ the total number of measurements of the sample. This analysis shows that 18 out of the 20 measurements are independent, implying that the signal decorrelates quickly after a few measurements. Based on the low autocorrelation, we can conclude that after approximately 3 measurements (around 30 days, as we have one measurement per cycle), the signal becomes largely decorrelated. In other words, measurements separated by more than one month have little to no correlation, meaning they can be treated as independent for analysis purposes.**

**We have better explain this method in the updated paper.**

L180 In this section you seem to focus on the Jason-3 and S6-MF tandem phases, but you do not clearly state that, you only mention that you use the duration of the first tandem phase from those missions and refer to Tab. 1 for the uncertainty budget, which contains the uncertainties for all 3 pairs of satellites. Could you please make this clearer.

**Corrected**

L193 10.16 seems to be the wrong number, probably a typo.

**Corrected**

L194-195 This conclusion comes out of nowhere because the specific Jason-3/S6-MF 2-tandem phase scenario has only been mentioned once before in the manuscript, at the end of the introduction, where the duration of the second phase is not mentioned at all, just when the second phase would be. You need to elaborate this scenario better before discussing results and conclusions about it, not after. It would also be good to know how and why was this particular scenario chosen.

**There is a misunderstanding here underlying the fact this section have to be improved. The conclusion "Following this analysis, a second tandem phase of 4 months is deemed sufficient to verify the instrumental stability on the global scale" is not related to the second tandem phase between S6 and Jason-3, but is derived from Fig. 3, which shows the evolution of the uncertainty of the trend of the GMSL differences as a function of the time period between the two tandem phases (between 1 and 6 years), for 4 different time spans of the second tandem phase (from 1 month to 6 months). The sentence has been reformulated to better understand the recommendation coming from the analysis presented in this paper.**

**The recommendations on the duration of the second tandem phase between S6 and Jason-3 are coming from preliminary results of the same study**

presented in conference (OSTST 2020) but not not yet published in a peer review paper.

Note that the duration of the second tandem phase between Jason-3 and S6-MF (4 months) has been added at the end of the introduction as recommended.

L199-206 You might want to put the sensitivity tests into a separate paragraph to improve readability of the manuscript.

**Done**

L212 I do not understand what does "corresponding to regional scales" mean in this context.

**Corrected: it is not relevant in this sentence, it has been removed.**

L221 Nothing is actually marked as a results section and there are two sections that describe the results, the previous one (Sect. 4) and this one (Sect. 5).

**Corrected: it has been replaced by "In this section"**

L228 These are not the same satellites as in Sect. 4. Are you comparing the results for the without-tandem method with the results from Sect. 4. for the Jason-3 and S6-MF or are you re-calculating the 2-tandem uncertainties for other satellites with different short-term time-correlated errors?

**Corrected. The uncertainty budget presented in Table B1 (from Jugier et al., 2022) is applicable to different altimetry missions.**

L232 & L240 You refer to Tab. B1 and Tab. C1 before Tab. A1.

**Corrected**

L285 methods, there are two of them

**Corrected**

L301-302 "since" is written twice

**Corrected**

Fig. 2. What is the unit of lag? When discussing the time with no autocorrelation, you use months, so could you explicitly relate that to whatever is shown here. Also, please note clearly what is the grey line in (f) here, not just in the main text.

**Corrected: the description of the grey line has been added to the figure description. The autocorrelation of the data is related to the time interval**

**between measurements, with one measurement taken per cycle (approximately every 10 days). It has been added to the x-label of autocorrelation plots.**

Fig. 3. Should be space agency, not spatial.

**Corrected**

Fig. 4. Since in the text you sometimes refer to box sizes in kilometers, could you please add that in the figure, it would make it easier to follow.

**Done**

Tab. 2. Could you please also provide the values for other box sizes used in the study?

**We provide here the value for other box sizes for information,but not added in the paper :**

- **Uσ = 4.1 mm for box size 3°*3°**
- **Uσ = 2.7 mm for box size 6°*6°**
- **Uσ = 2.3 mm for box size 9°*9°**
- **Uσ = 2.1 mm for box size 12°*12°**
- **Uσ = 1.9 mm for box size 18°*18°**
- **Uσ = 1.6 mm for box size 36°*36°**

There is an error with the format of citations throughout the paper, most of them have too many brackets.
**Done**

---

## Author Response (AR2)

**Report #1**

Submitted on 05 Nov 2024
Anonymous referee #1

**Anonymous during peer-review: Yes** No
**Anonymous in acknowledgements of published article: Yes** No

**Checklist for reviewers**

| | |
|---|---|
| **1) Scientific significance**
Does the manuscript represent a substantial contribution to scientific progress within the scope of this journal (substantial new concepts, ideas, methods, or data)? | **Excellent** Good Fair Poor |
| **2) Scientific quality**
Are the scientific approach and applied methods valid? Are the results discussed in an appropriate and balanced way (consideration of related work, including appropriate references)? | Excellent **Good** Fair Poor |
| **3) Presentation quality**
Are the scientific results and conclusions presented in a clear, concise, and well structured way (number and quality of figures/tables, appropriate use of English language)? | Excellent **Good** Fair Poor |

**For final publication, the manuscript should be**

accepted as is

**accepted subject to technical corrections**

accepted subject to **minor revisions**

reconsidered after **major revisions**

rejected

**Were a revised manuscript to be sent for another round of reviews:**

**I would be willing to review the revised manuscript.**

I would not be willing to review the revised manuscript.

**Report #2**

Submitted on 14 Nov 2024
Anonymous referee #2

**Anonymous during peer-review: Yes** No
**Anonymous in acknowledgements of published article: Yes** No

**Checklist for reviewers**

| | |
|---|---|
| **1) Scientific significance**
Does the manuscript represent a substantial contribution to scientific progress within the scope of this journal (substantial new concepts, ideas, methods, or data)? | Excellent **Good** Fair Poor |
| **2) Scientific quality**
Are the scientific approach and applied methods valid? Are the results discussed in an appropriate and balanced way (consideration of related work, including appropriate references)? | Excellent **Good** Fair Poor |
| **3) Presentation quality**
Are the scientific results and conclusions presented in a clear, concise, and well structured way (number and quality of figures/tables, appropriate use of English language)? | Excellent **Good** Fair Poor |

**For final publication, the manuscript should be**

**accepted as is**

accepted subject to **technical corrections**

accepted subject to **minor revisions**

reconsidered after **major revisions**

rejected

**Were a revised manuscript to be sent for another round of reviews:**

**I would be willing to review the revised manuscript.**

I would not be willing to review the revised manuscript.

**Suggestions for revision or reasons for rejection**
(visible to the public if the article is accepted and published)

The authors have answered all my comments, I recommend publication of this work.

**Suggestions for revision or reasons for rejection**

(visible to the public if the article is accepted and published)

Second review of "Benefits of a second tandem flight phase between two successive satellite altimetry missions for assessing the instrumental stability" by Ablain et al. The authors present a validation method to assess the instrumental drifts in the global and regional mean sea level trends based on the implementation of a second tandem phase between two successive satellites. The topic is a significant scientific contribution and, after the authors implemented all the changes to the manuscript, the methods and the results are valid and presented in a clear and well structured way. There are still a few small and mainly technical things that should be corrected, but other than that I recommend accepting this manuscript.

Here is a list of corrections that should still be made in order of appearance:

L55, L70-71 There should be no parentheses around the year.
Corrected

L103 It would be better to use the word line instead of curve. Curve implies that a line is curved, and this one is not.
Corrected

L130 tree -> three
Corrected

L167 There is a citation missing.
Maybe a problem when the pdf file was exported, because the citation is not missing.

L195 space missing
Corrected

L203 space missing
Corrected

L268, L272 There seem to be extra spaces between the opening brackets and the numbers.
Corrected

L278 extra space
Corrected

L296 extra space twice
Corrected

L297-298 It does not really make sense to write "as a function of the time spent between... (2 years and 9 months)". Either it is a function of time, or it is for a fixed time of 2 years and 9 months. Perhaps you would like to say something like: "as a function of time..., with special focus on 2 years and 9 months as in the adopted scenario"?
Corrected by removing the text "(2 years and 9 months)" which is not useful in this sentence.

L341 This is the only time in the manuscript you refer to Jason-3 as J3, except in the figures. Since you use the whole name everywhere else, it would be better to use it here too (or to use the abbreviation throughout the manuscript).
Corrected

Fig 2 For (c) caption says J3/S6-MF, but in the figure it is J3/S6A
Corrected

Figs 3-6 have "Uncertainties at 68 % CL (1sigma)" in the legend. That is rather confusing, especially in Fig 6, where you also show the envelopes. It would be better if you moved this to the y axis label (deltaRMSE trend uncertainty at 68 % CL...).
Corrected

Fig 4 There should be no space between the degree symbol and the number.
Corrected

Fig 6 It says differences twice in the caption.
Corrected

Table 2 Thank you for providing those uncertainties in the reply, but it would be better to also have them in the table. In Table 1 you provide values for all three mission pairs, even though you only use one, so it is strange that here you only provide the values for one grid box size, even though you use all of them.
Corrected